# Endoprosthetic Reconstruction in Ewing’s Sarcoma Patients: A Systematic Review of Postoperative Complications and Functional Outcomes

**DOI:** 10.3390/jcm11154612

**Published:** 2022-08-08

**Authors:** Jude Abu El Afieh, Marena Gray, Matthew Seah, Wasim Khan

**Affiliations:** Division of Trauma & Orthopaedic Surgery, Department of Surgery, Addenbrooke’s Hospital, University of Cambridge, Cambridge CB2 0QQ, UK

**Keywords:** Ewing’s sarcoma, prostheses, limb salvage, surgery, complications, functional outcome, failure mode

## Abstract

Ewing’s sarcoma (ES) is a rare primary bone cancer managed by radiotherapy, chemotherapy and surgical resection. The existing literature on limb salvage surgery with endoprostheses combines data for ES patients with osteosarcoma. This review aimed to evaluate surgical and functional outcomes of endoprosthetic reconstruction in exclusively Ewing’s patients. We believe that this is the first comprehensive review to evaluate the outcomes of limb salvage surgery with endoprostheses exclusively in Ewing’s sarcoma patients. Clinical data and outcomes were collected from PubMed, Embase, Medline and Scopus. The inclusion criteria were studies on limb salvage surgery in ES patients, where individual patient data was available. Seventeen studies with a total of 57 Ewing’s patients were included in this review. Fifty-three of the ES patients preserved the limb after limb salvage with endoprostheses. The average five-year implant survivorship was 85.9% based on four studies in this review. Postoperative complications were categorised by Henderson’s failure modes. Soft tissue failure was the most common, occurring in 35.1% of patients, followed by deep infection in 15.7% of patients. There was a suggestion of ‘good’ functional outcomes with limb salvage surgery. The salient limitation of this review is the variability and rarity of the patient population. Homogenous data in a larger population is necessary to provide more insight into outcomes of limb reconstruction in ES.

## 1. Introduction

Ewing’s sarcomas (ES) are the second most common primary bone cancer in young people, after osteosarcoma [1]. Typically presenting with bone pain and a local mass, the common sites of primary involvement include the long bones of the lower limb and the bones of the pelvis and chest wall. Imaging with plain radiographs demonstrates a classically layered destructive, periosteal calcification [1,2]. The literature has demonstrated no inherited component to ES, unlike osteosarcomas in which 3% are linked to a mutation in p53, however in 95% of cases a t(11;22)(q24;q12) translocation is detected [1,3]. 

Management of ES is achieved with a combination of systemic chemotherapy and local control, typically through surgical resection [1]. ES and other Ewing’s sarcoma family of tumours (ESFT) are also quite responsive to radiotherapy unlike osteosarcoma, which rely on neo-adjuvant and adjuvant chemotherapy following resection [3]. Resection can involve the physis, which poses the risk of limb-length discrepancy in the skeletally immature child [2]. In order to match the growth of the healthy limb, the development of expanding prostheses was necessary and has revolutionised limb salvage, reducing the need to resort to amputation or rotationplasty [1,4]. Limb salvage surgery with prosthesis can lead to a variety of potential postoperative complications. Henderson et al. developed a classification system for types of failure of limb salvage following endoprosthetic reconstruction [5]. The orthopedic surgery component therefore must balance aiming for the best survival outcomes without jeopardising patients’ limbs and mobility entirely. 

Due to the similarities in presentation and management, many studies combine ES and osteosarcoma when reviewing the outcomes of therapeutic interventions or surgeries. However, ES is known to be susceptible to radiotherapy, and the treatment regimen for localised ES has frequently included radiotherapy primarily. There has been a decline in radiotherapy being used as a definitive treatment method in the last decade as this approach has been linked with complications, such as aseptic loosening, infection, delayed wound healing and adverse effects on bone growth, particularly when it is used in addition to chemotherapy [6,7,8,9,10]. The chemotherapy regimens for the management of ES and osteosarcoma also vary in nature and duration. For example, all current ES management trials (local disease) involved 10 to 12 months of treatment comprising multiple cycles of neoadjuvant chemotherapy, followed by local therapy and further cycles of chemotherapy [11]. By comparison, the chemotherapy drugs used vary and the chemotherapy duration for OS is typically shorter [11]. In addition to this, ES is almost exclusively high grade and typically involves the diaphyseal region of bones, whereas there is more heterogeneity in osteosarcoma, which usually involves the metaphyseal region [1,2]. Therefore, the overall aim of this study was to systematically review and evaluate the complications which occur in reconstructive surgery of Ewing’s sarcoma patients only. There are two objectives: (i) to investigate postoperative outcomes, surgical and functional, for reconstructive surgery with prosthesis in ES patients and (ii) determine which prosthesis offers the best outcomes for patients with ES. Due to the lack of previous studies looking at the endoprosthetic reconstruction in ES only, we will include all types of prosthesis as part of the initial search strategy. We believe that this is the first comprehensive systematic review to evaluate the outcomes of limb salvage surgery with endoprostheses exclusively in Ewing’s sarcoma patients. 

## 2. Materials and Methods

### 2.1. Search Strategy

The authors performed an online computed-based review of published literature from the following databases: PubMed, Embase, Medline and Scopus, in accordance with the preferred reporting items for systematic review and meta-analysis (PRISMA) protocols [12]. The following terms used for the search were ‘Ewing’s Sarcoma’ and ‘prostheses’. All databases were searched from their inception until 20 February 2022. 

### 2.2. Selection Criteria

The titles, abstracts and results of the studies were reviewed by the primary authors, JAA and MG, who used the predefined criteria to choose the most relevant publications. Data collected included number of cases recorded, demographic characteristics, the type of prosthesis and post-surgical complications, as well as follow-up duration and postoperative functional score. The following are the inclusion criteria for this review: article was published in a peer reviewed journal in English, full-text publication and described a sample of at least one patient with ES. Exclusion criteria included: figures which combined Ewing’s and osteosarcoma cases and complications, studies which did not include type of prosthesis or complications, ES of the soft tissue and studies where the tumour does not originate from the long bones. The search algorithm is illustrated in Figure 1. 

### 2.3. Data Collection

All data was entered into a spreadsheet and primarily categorised according to the study, location of the Ewing’s tumour and the type of prosthesis used. The age, sex, follow-up length, postoperative complications and functional score were also examined. Five models of prosthesis failure types were identified and classified as per Henderson’s model: soft tissue failure (Type 1), aseptic loosening (Type 2), structural failure (Type 3), infection (Type 4) and local tumour progression (Type 5) [5]. 

### 2.4. Quality Assessment 

The Newcastle-Ottawa Scale was used to assess individual study quality [13]. The scale judges each study in three categories: selection of the study group, comparability of the group, and assessment of outcome. The maximum stars possible for the involved criteria are 4 in selection, 2 in comparability and 3 in outcome. 

## 3. Results

### 3.1. Search Results

The literature search resulted in a total number of 811 publications. After duplicates were removed and an initial review based on title and abstract was done, 80 publications remained. Further exclusion criteria were applied, and content analysis resulted in the exclusion of 63 publications for the following reasons: not involving long bones, not discussing the complications of the procedure, and not separating the data of Ewing’s and osteosarcoma patients. Seventeen publications remained. The chosen studies were assessed for quality, a total of 5 to 9 stars identifies a study to be of high methodological quality [13]. All chosen studies were deemed to be high quality, scoring at least a six on the Newcastle-Ottawa Scale (Table 1).

### 3.2. Design and Content of the Studies

The 17 included studies were published between 1986 and 2021. The studies included each followed a single cohort of patients who received an intervention of reconstructive surgery with prosthesis. After excluding non-Ewing’s patients from these studies, 57 patients who had limb reconstruction using prosthesis were identified. Two of the studies did not report the sex of their patients [14,20]. The mean age of the patients at the time of their surgery was 14.5 years (range 4–46 years). The most common anatomical site of Ewing’s sarcoma was the proximal humerus, followed by the proximal femur, distal femur and distal tibia. In three studies, the cancer was in the femoral diaphysis but the exact location was not specified [17,26,30]. The most common type of prosthesis used in these studies was Stanmore, which was used in 23 patients in seven studies. Other prostheses included were Repiphysis, MUTARS, Kotz, LUMiC, RESTOR, Lewis, HMRS and Non Hinged CCK megaprosthesis (Table 2). Six studies used non-expanding prostheses [17,18,22,25,26,29], while the remaining eleven studies involved expanding prostheses [14,15,16,19,20,21,23,24,27,28,30]. Some studies used an expanding Stanmore prosthesis; however, three studies used custom made non-expanding Stanmore implants [17,18,22]. 

### 3.3. Postoperative Complications

For the purposes of this systematic review, complication is defined as side effects of prosthesis that require intervention. For the included studies, failure modes were described based on the classification by Henderson et al. [5]. Among the 57 patients, forty one complications and failure modes were reported in ten studies (Table 2). 

The most common failure mode was found to be Type I failure. This occurred in nineteen patients and included complications, such as restricted range of motion, wound dehiscence, joint instability (including dislocations), superficial infections and nerve lesions. Restricted range of motion was usually due to contractures, which occurred in six patients. Joint instability (including dislocation) occurred in five patients. Superficial infections, nerve injury and wound healing complications, specifically wound dehiscence, occurred in a total of eight patients. 

Deep infection (Type IV failure mode) was also a common failure mode, seen in nine patients in five studies (Table 2). Only one of the included studies investigated the responsible pathogens. They were able to isolate the following bacteria: Staphylococcus aureus, Pseudomonas species, Enterococcus species and Escherichia coli [20]. 

Type II failure, aseptic loosening, occurred in three patients from the same study [20]. This was described as a late complication, as it occurred more than 1 month after surgery. Type III failure is described as structural failure and encompasses prosthetic failure and periprosthetic fracture. Type V failure is categorised by tumour progression or recurrence with endoprosthesis contamination [5]. This occurred in three patients who consequently had an amputation. There was no significant difference in complication rates between expandable and traditional prostheses. 

### 3.4. Prosthesis Survivorship 

The five-year prosthesis survivorship was analysed in four of the selected studies and was based on Kaplan-Meier curves [17,25,26,28]. The starting point was defined as the date of implantation of the prosthesis and the endpoint being when it was removed regardless of what the cause was. The mean five-year survivorship of the implant based on these four studies was 85.9%, with a range of 80.9–90%. This was for Stanmore, LUMiC and RESTOR prostheses. The remaining thirteen studies did not consider implant survival; however, the range of follow-up time for the patients post implantation in these studies was 6–324 months. Nevertheless, due to the limited sample size and inability to extrapolate this data for exclusively Ewing’s patients, definite conclusions on prosthesis survival were not drawn. 

### 3.5. Functional Outcomes

Functional outcomes of Ewing’s patients were assessed in fourteen studies. The Musculoskeletal Tumour Society score (MSTS) questionnaire was the scale that was most commonly used to assess functional outcome. This was used in ten of the chosen studies across 29 patients (Table 2). The mean MSTS score across 29 patients was 77.5%, comparing Repiphysis, MUTARS, Stanmore, LUMiC, RESTOR and non-hinged CCK implants. Two studies used the Enneking score to assess functional outcomes in Stanmore and Kotz prostheses with an average score of 77.9% [24,25]. The International Symposium on Limb Salvage (ISOLS) scale was used for a single patient in a study by Yoshida et al. [21], while AMSTS, a qualitative scoring system, was used for six patients in a study by Dotan et al. [20]. Several cases in two studies had their functional score omitted due to inadequate length of follow-up [20,27]. 

## 4. Discussion

In the past, most patients with ES would have had an amputation of the limb involved, leading to poor functional outcomes [31]. Endoprosthetic reconstruction continues to be widely used as an alternative to amputation in the treatment of malignant bone tumours. The main objective of limb salvage surgery is to help ensure patients are able to have near normal function of their limb as soon as possible, without compromising survival [32]. Expandable and adjustable internal prostheses promised an improved cosmetic result and immediate weight bearing [33,34]. They also helped overcome the problem of anticipated limb length discrepancy, especially in pediatric patients [30]. 

### 4.1. Prostheses Comparison 

The most common prostheses used in the included studies were Stanmore in 23 patients and Kotz in 15 patients. Overall, there were 23 complications recorded for ES patients who underwent limb salvage surgery with Kotz prostheses. Type I failure type was the most common failure mode reported, with 66.7% of patients developing complications, such as superficial infection, contracture, wound dehiscence and dislocation. Comparatively, there were seven complications in 23 patients who had limb salvage surgery with Stanmore prosthesis, with Type I failure mode also being the most common, reported in 21.7% of patients. There are major points to consider when comparing these prostheses. Firstly, in the included studies, Kotz was more commonly used for tumours originating in the proximal femur, while Stanmore was most commonly used for proximal humerus tumours. Additionally, most of the data extrapolated regarding Kotz prostheses originated from a single study by Dotan, which included 12 of the 15 patients [20]. It was also difficult to compare functional scores for Kotz prosthesis compared to Stanmore as Dotan et al.’s study used a qualitative scoring system, AMSTS, to evaluate the patients’ functional outcomes [20]. Some studies using Stanmore prosthesis used MSTS to evaluate functional outcome [17,18,22,28], while others used Enneking [24,27]. Due to the aforementioned variables, as well as differences in non-surgical treatment received and follow-up duration, we were unable to draw definitive conclusions on which prosthesis is associated with better functional outcome and less post-operative complications. 

Expandable prostheses (e.g., Repiphysis and Stanmore) were developed to preserve limb function while maintaining growth following reconstructive surgery involving the physis. These implants underwent a series of generational improvements to reduce morbidity related to lengthening and to improve prosthesis survival. Newer models can be lengthened ‘non-invasively’ without the need for surgery or anaesthesia. Mechanical failure was reported in 2 patients (Kotz prosthesis) [20]. However, due to the small numbers of ES patients reported in the studies, no conclusions were drawn with regard to the type of prosthesis and the incidence of specific complications. 

### 4.2. Outcomes 

Postoperatively, complications were reported in this review, with Type I failures reported as the most common. Dislocations were managed with manipulation under anesthesia and immobilisation in plaster in one case [15], and with open reduction and soft tissue reconstruction in the other cases [20,25]. Reduced range of motion is established as a prevalent complication in expanding prostheses, as it is associated with multiple expansions of the prosthesis [19]. Intensive postoperative physiotherapy is usually the initial approach for treating contracture. However, the six patients who developed contractures postoperatively in this review had surgical release to manage this, which resulted in good functional range of motion [14,20]. Contracture has been described in existing literature as a radiation-associated complication that can occur in patients who received radiotherapy for ES [35]. More research needs to be done to investigate whether contracture is more prevalent in ES patients compared to other sarcomas, as Ewing’s tumours are responsive to radiotherapy and tend to be treated with this preoperatively [36]. Another concern regarding radiotherapy as treatment in ES is secondary malignancy. The Italian Sarcoma Group Experience reviewed the late effects of radiotherapy in ES patients over 23 years and found that 2.8% of patients developed a secondary malignancy, most commonly radiation-induced osteosarcoma [37]. Moreover, a report that analysed the German Ewing’s Sarcoma Studies CESS 81 and CESS 86 found a 4.7% incidence of secondary malignancy over 15 years in ES patients [38]. 

Aseptic loosening of prosthesis, type II failure, is a disabling complication which can occur years after the initial surgery, usually as a result of inadequate initial fixation of the implant, mechanical loss of fixation over time or biological loss of fixation caused by osteolysis [39]. All three patients who suffered from this complication were treated with revision surgery [20]. 

Type III failure includes mechanical failure and periprosthetic fractures. Mechanical failure occurred in two patients with Kotz prosthesis. These were late complications and did not have a connection to the elongation of the prosthesis. Both were treated with revision surgery. One periprosthetic fracture was reported, and was managed conservatively with plaster of Paris and a period of non-weight bearing [20]. 

Deep infection (Type IV failure) was a common complication seen in 15.7% of the included patients, likely due to the immunocompromised status of the patient cohort [40]. A study by Jeys et al. described a periprosthetic infection rate of approximately 18% in limb salvage with expandable endoprosthesis, and that infection was increased by 5% each time a patient underwent a lengthening procedure [41]. The most common cause of these infections was found to be coagulase-negative staphylococcus. In one patient, deep infection was managed with washout, debridement and implant retention [28]. The remaining patients with this complication received revision surgery with implant replacement [14,17,19,20]. It has been suggested that there is a higher risk of infection after endoprosthetic tibial reconstruction due to difficulty in achieving soft tissue cover, compared to tumors located in the femur diaphysis [42]. There is literature available on preventative measures that can be taken to lower the risk of deep infection involving endoprosthesis. A study by Hardes et al. examined the infection rate in patients with sarcoma in the proximal tibia who underwent limb salvage surgery with silver- or titanium-coated megaprosthesis. Infection rate in the titanium group was 16.7% compared to 8.9% in the silver group, suggesting that the use of silver coated prosthesis reduced infection rate [43]. There has been interest in establishing guidelines regarding use of perioperative antibiotic prophylaxis limb salvage surgery for sarcoma patients [44]. Opinions on this matter vary widely, suggesting the need for a randomised trial to investigate this and ultimately aid the development of guidelines. Further work should investigate measures to minimise the risk of deep infection, so as to reduce the need for further revisions and implant replacement in vulnerable patients. 

In this review, 53 patients preserved the limb after limb salvage with endoprostheses. Three patients in this review had type V failure mode and had an amputation due to local spread of disease [20,21,22]. A fourth patient also had an amputation, but this was due to postoperative pain that could not be managed conservatively [27]. 

Formal five-year implant survivorship data was calculated as an average of 85.9% based on only four studies in this review. It is imperative to note that this data is based on the survival of prostheses in patients with a multitude of different primary bone tumours who had limb salvage surgery, and may not be representative of prostheses survival in ES patients. Follow-up time scales were variable and inconsistent with and between studies, with patients lost to follow-up before information regarding implant survival could be established and other patients dying of their disease before the implant failed. 

Functional outcomes were determined by the Musculoskeletal Tumour Society Score questionnaire in the majority of studies in this review. This scoring system included common domains such as pain, function and emotional acceptance. It also consists of extremity specific domains [45]. Erol et al. demonstrated improvement in MSTS before and after surgery, with an average of 33.9% improvement in the functional score postoperatively, thus further demonstrating the benefits of limb salvage surgery [25]. Across the studies, while there is the suggestion of ‘good’ functional scores (Table 2), it is difficult to state for certain the significance of these functional outcomes, given the small number of cases and high proportion of those with no data. 

### 4.3. Limitations 

The salient limitation of this review is the variability of the patient population, as only four of the selected studies included more than five Ewing’s sarcoma patients. The small subset of patients included is unlikely to be a representative sample of the total Ewing’s sarcoma patient group, and thus results of this review should be interpreted with caution. We have noted the location of the Ewing’s tumour in each case in Appendix A; however, due to the heterogenous nature of the data, we could not isolate whether certain complications were associated with specific anatomical regions. 

Additionally, there was variability of the initial non-surgical management of these patients, location of the Ewing’s tumour and type of implant used. We attempted to consider the variation in the nature of the resections and adjuvant treatments used but due to the heterogeneity of patients’ treatment profiles, and lack of detail about this in some of the involved studies, we could not draw meaningful conclusions. The details found on non-surgical treatment in these patients has been provided in the Appendix A.

Follow-up duration of the included studies was varied and sometimes incomplete. The existing literature shows that the incidence of implant-related complications increases with longer follow-up duration for osteosarcoma patients, but no current data concludes this in Ewing’s patients [46]. Case reports were also excluded from our review, so new advances in endoprosthetic reconstruction in ES may not have been considered. In spite of these limitations, an initial evaluation of postoperative outcomes of limb salvage surgery in ES patients was conducted. We believe that this review contributes to the minimal existing literature for this group of patients. 

## 5. Conclusions

To conclude, when planning for endoprosthetic reconstruction in ES patients, it is imperative to consider a multitude of factors, including long-term prostheses survival, risk of complications and functional outcomes. Soft tissue failure and deep infection were the most common complications we found, and are associated with lengthening procedures of expandable prostheses. Ultimately, the goal is limb salvage through noninvasive endoprosthesis, as well as acceptable functional outcomes, without jeopardising the eradication of Ewing’s tumour. Homogenous data in a larger population is necessary to provide insight into specific outcomes of limb reconstruction in ES. This could particularly help determine a superior prosthesis type to be used in ES patients and thus lead to improved post-surgical and functional results. Large single institutions or collaborative trials that are in possession of comparable data on ES patients should be encouraged to publish the data already available to them. If this data is not available, researchers should consider data sampling ES patients in future data collection as this could potentially pave the way for the development of novel prostheses, surgical techniques and postoperative care in this rarely represented group of patients.

## Figures and Tables

**Figure 1 jcm-11-04612-f001:**
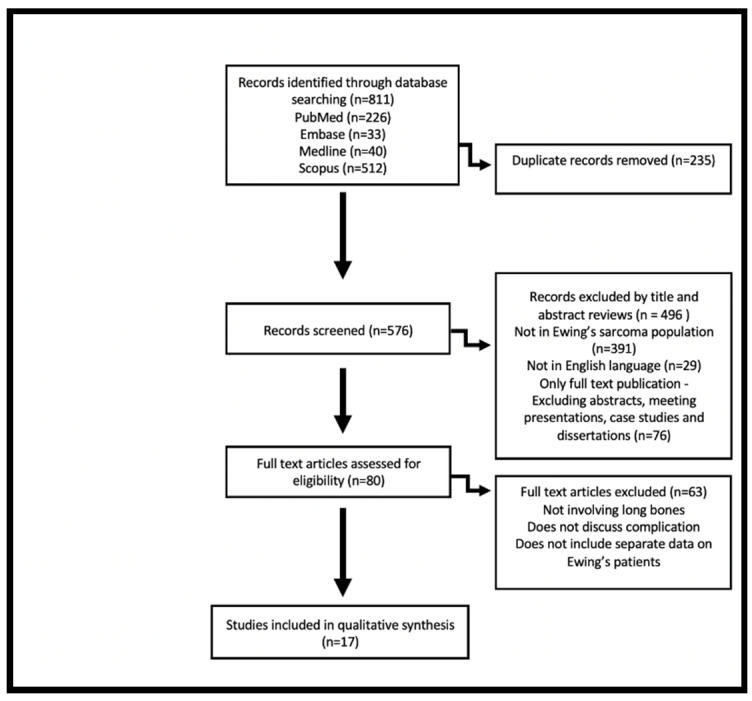
Flow diagram visually summarising results of literature search and screening process based on Preferred Reporting Items for Systematic Reviews and Meta-analysis (PRISMA) guidelines [12].

**Table 1 jcm-11-04612-t001:** Newcastle-Ottawa quality assessment of included studies. The maximum stars possible for the involved criteria are 1 in selection, 2 in comparability and 3 in outcome. * AHRQ: Agency for Healthcare Research and Quality.

Author, Year	Selection	Comparability	Outcome	Quality Based on AHRQ *
Benevenia 2015 [14]	★★	★	★★★	Fair
Vijayan 2011 [15]	★★★	★★	★★★	Good
Torner 2016 [16]	★★★	★★	★★★	Good
Hanna 2010 [17]	★★★	★	★★★	Good
Yang 2017 [18]	★★★	★	★★★	Good
Schiller 1995 [19]	★★★	★★	★★★	Good
Dotan 2010 [20]	★★★	★	★★★	Good
Yoshida 2011 [21]	★★★	★	★★★	Good
Shekkeris 2009 [22]	★★★	★	★★★	Good
Raciborska 2021 [23]	★★★	★★	★★★	Good
Yoshida 2008 [24]	★★★	★★	★★★	Good
Erol 2021 [25]	★★★	★	★★★	Good
Puri 2012 [26]	★★★	★	★★	Fair
Ayoub 1999 [27]	★★★	★★	★★	Good
Wafa 2015 [28]	★★★	★	★★★	Good
Ji 2019 [29]	★★★	★★	★★★	Good
Lewis 1986 [30]	★★★	★	★★★	Good

**Table 2 jcm-11-04612-t002:** Summary of extracted date of patient characteristics prosthesis type and complications from included studies.

Author, Year	N	Age Range	Prosthesis	Complications	Mean FS
Benevenia, 2015 [14]	2	11–16	Repiphysis	Contracture×2,	95%(MSTS)
				Deep Infection	
Vijayan, 2011 [15]	1	4	Stanmore	Joint instability, flexion deformities	-
Torner, 2016 [16]	1	12	MUTARS	-	86% (MSTS)
Hanna, 2010 [17]	3	10–25	Stanmore	Deep infection	79.7% (MSTS)
Yang, 2017 [18]	1	26	Stanmore	-	67% (MSTS)
Schiller, 1995 [19]	1	9	HMRS	Deep infection, joint instability	-
Dotan, 2010 [20]	12	6–14	Kotz	Superficial infection ×4, Deep infection ×5, Contracture ×3, Mechanical failure ×2, Wound dehiscence, Dislocation ×2, Aseptic Loosening ×3, Periprosthetic fracture, Amputation due to local recurrence	Good (AMSTS)
Yoshida, 2011 [21]	1	12	Kotz	Amputation due to local recurrence	88% (ISOLS)
Shekkeris, 2009 [22]	2	15–42	Stanmore	-	76.5% (MSTS)
Raciborska, 2021 [23]	4	6–18	MUTARS	-	73% (MSTS)
Yoshida, 2008 [24]	3	7–12	Stanmore + Kotz	-	73.6% (Enneking)
Erol, 2021 [25]	6	17–46	LUMiC	Amputation due to local recurrence, Dislocation	67.7% (MSTS)
Puri, 2012 [26]	1	17	RESTOR	-	80% (MSTS)
Ayoub, 1999 [27]	8	6–11	Stanmore	Wound dehiscence,	80% (Enneking)
				Radial nerve palsy, Amputation for pain	
Wafa, 2015 [28]	7	7–32	Stanmore	Superficial wound infection, Periprosthetic infection	82.9% (MSTS)
Ji, 2019 [29]	2	9–10	Non-hinged CCK	-	77% (MSTS)
Lewis, 1986 [30]	2	10–13	Lewis	-	-

**FS:** Functional score, **MSTS:** Musculoskeletal Tumour Society score, **ISOLS:** International Society of Limb Salvage score, **AMSTS:** American Musculoskeletal Tumour Society score, **Enneking:** Enneking’s functional evaluation.

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
