# Peer review of "Endoprosthetic Reconstruction in Ewing’s Sarcoma Patients: A Systematic Review of Postoperative Complications and Functional Outcomes"

_jcm, 2022, doi:10.3390/jcm11154612_

Round 1
Reviewer 1 Report
Thank you to the authors for the subject of this review and their work trying to find some valid and useful information regarding the functional outcomes and complications after endoprosthetic reconstructions in Ewing Sarcomas;
Unfortunately, the number of cases reported in literature and data quality are still major obstacles to solid conclusions that might guide our orthopedic treatments in these cases. The authors have shown and repeatedly highlighted this issue along their manuscript. Nonetheless, I think these review add some important data into the current knowledge within this specific field. For this reason I believe this manuscript deserves to be published.
However, there are some minor corrections, which should be done before the final publication:
1. Line 43 - Where we can read "pain radiographs" please correct to "plain radiographs"
2. Lines 46 and 47 - I believe the sentence "It should define the purpose of the work and it significance" does not make sense there - please verify
3. Line 174 - please revise the sentence "Functional outcomes were of Ewing's patients were..."
4. Line 224 - Please revise "Complications post-operatively" to "Post-operatively complications" (I believe It sounds better)
5. Line 271 - Please correct "A fourths patients" to "A fourths patient.."
Thank you for the opportunity to revise the manuscript
Best Regards
Reviewer 2 Report
The authors have a done a good job in identifying the previously published literature on this topic. Although firm conclusions cannot be drawn from this review it serves as the benchmark for future studies.
As the authors identify that use of radiation therapy is one of the key differences between ES and OS management, then what were their findings in their review that are attributed to its use? This should be described in detail.
Apart from the use of radiation what are other reasons for exclusively studying ES patient outcomes?
